# Gender Differences in Factors Influencing Self-Efficacy Toward Pregnancy Planning among College Students in Korea

**DOI:** 10.3390/ijerph17103735

**Published:** 2020-05-25

**Authors:** Saem Yi Kang, Hae Won Kim

**Affiliations:** 1College of Nursing, Seoul National University, Seoul 03080, Korea; rkdtoadl@snu.ac.kr; 2Research Institute of Nursing Science, College of Nursing, Seoul National University, Seoul 03080, Korea

**Keywords:** pregnancy, preconception care, gender

## Abstract

The purpose of this study was to examine factors influencing college students’ self-efficacy toward pregnancy planning by gender using the health belief model (HBM). Utilizing a comparative descriptive design, a total of 819 college students were recruited. A survey was administered to gather information on health beliefs related to pregnancy planning, self-efficacy toward pregnancy planning, fertility knowledge, and general characteristics. The main variables were compared by gender. The factors influencing self-efficacy toward pregnancy planning were identified using hierarchical regression analysis. Female students (476) had lower self-efficacy toward pregnancy planning than male students (343). The significant factors influencing self-efficacy toward pregnancy planning in female students were: depression (β = −0.09, *p* = 0.030), fertility knowledge (β = 0.08, *p* = 0.025), barriers (β = −0.57, *p* < 0.001), and cues to action (β = 0.16, *p* < 0.001), whereas the corresponding factors in male students were benefits (β = 0.12, *p* = 0.020), barriers (β = −0.44, *p* < 0.001), and cues to action (β = 0.16, *p* = 0.001). The present study confirmed the suitability of the HBM as a conceptual framework for identifying factors influencing self-efficacy toward pregnancy planning. Based on the findings of this study, gender-based similarities and differences in factors influencing self-efficacy should be considered when taking steps to promote self-efficacy toward pregnancy planning among college students.

## 1. Introduction

Pregnancy planning is the first step of preconception care as set out by the Centers for Disease Control and Prevention [1]. Previous studies have demonstrated that women who planned to become pregnant had more favorable pre-pregnancy health behaviors such as folic acid intake, smoking cessation, and alcohol abstinence [2,3,4], whereas unplanned pregnancies increased the likelihood of abortion miscarriage, delayed or missed antenatal care, and engaging in unhealthy behavior during pregnancy [5,6,7]. Furthermore, men planning for pregnancy made lifestyle changes and managed their health more closely before pregnancy, showing improvements in sperm quality [8]. Even before marriage, young adults can improve their pre-pregnancy health behaviors by planning for future pregnancy and childbirth [9]. Therefore, college students should consider pregnancy planning to maximize their likelihood of a healthy pregnancy outcome.

Especially for college students who are entering adulthood and becoming more sexually active, the concept of self-efficacy provides an effective framework for measuring their motivations to practice pregnancy planning instead of their current health actions. Therefore, this study was designed to measure self-efficacy because it is linked to continued motivation for future pregnancy planning. Self-efficacy has been added as a variable to the health belief model (HBM) to enhance its explanatory power [9]. A recent study reported that self-efficacy regarding physical activity for pregnancy-related weight management was significantly explained by using the HBM and significant correlations were found between self-efficacy, and other health belief variables [10]. Based on those previous studies, within the framework of the HBM, thoughts about pregnancy planning among unmarried young adults could be explained in terms of self-efficacy and health beliefs, among which associations were postulated.

Fertility knowledge has been shown to have a significant influence on family planning for women [11] and lack of fertility knowledge has been found to impede preconception care for men [12]. Furthermore, young adults’ fertility knowledge has been reported to be low in general [13], making it necessary to assess fertility knowledge among Korean college students and to determine whether it affects self-efficacy toward pregnancy planning. Pregnancy planning has also been found to be associated with maternal characteristics such as age, income, smoking, alcohol drinking, stress, and depression [14,15,16]. Sexual experience also showed a significant relationship with reproductive health-promoting behaviors among college students [17]. In light of those findings, the present study sought to determine whether these variables influenced self-efficacy toward pregnancy planning.

Traditionally, women play a greater biological or social role as mothers than men in relation to childbirth [18]. Even before pregnancy, women have a stronger belief than men that they should receive preconception care [19]. Although preconception care is essential for both women and men [20], a systematic literature review showed that only a few studies included men as subjects [21]. A survey of pregnant women in Korea reported that about half of the respondents had not planned for pregnancy [7]. Moreover, the frequency of pregnancy planning among men has not been investigated. Therefore, this study approached pregnancy planning from a gender perspective.

In summary, it is important to start preconception care early to maximize the likelihood of healthy pregnancy outcomes and self-efficacy can be used as a theoretical construct for explaining future thoughts about pregnancy planning among college students. Through a comparative approach between men and women, pregnancy planning can be recognized as an important aspect of reproductive health regardless of gender. The aim of this study was to compare gender differences in self-efficacy and beliefs toward pregnancy planning in the framework of the HBM. We also assessed the factors influencing self-efficacy toward pregnancy planning by gender.

## 2. Materials and Methods

### 2.1. Study Design and Sample

We conducted a cross-sectional survey of college students at Seoul National University (SNU) in South Korea. The study participants were college students who used their school email addresses. The inclusion criteria required participants to be between 18 and 30 years of age and unmarried. In total, 1273 college students participated in the study. After excluding incomplete responses and questionnaires that lacked consistency, the final analyzed sample consisted of 819 students (476 women, 343 men). The participants with missing data on inconsistent responses for some questionnaire items (84) showed no significant differences in any general characteristics from the final analyzed sample.

### 2.2. Procedure

The research protocol of the project was approved by the SNU Ethics Committee (No. 1906/002-012). This project consists of a questionnaire with two main parts and the present study was conducted to investigate university students’ perceptions of pregnancy planning as the first part of the larger project. The goals of the larger project were to examine gender differences on awareness related pregnancy and birth among young adults. Data were collected from 20 July 2019 to 16 September 2019. All students were sent an email informing them about the survey with a link to the questionnaire via Survey Monkey, an online survey website. Before participants started the survey, a thorough description of the research was provided on the first screen, and the survey only started if the respondent agreed to participate in the study. In order to prevent duplicate participation and to ensure reliability of data, the IP address of the system itself and the mobile phone number entered for the payment of participants were checked. After completing the survey, the participants were provided a voucher (US $4) as a token of appreciation for completing the survey.

### 2.3. Measurement

The content and constructs of the developed questionnaire were validated by five experts in nursing and women’s health nursing.

#### 2.3.1. Health Belief Variables about Pregnancy Planning

We developed a questionnaire on health beliefs regarding pregnancy planning by combining newly developed HBM questions with self-efficacy variables. For questions dealing with perceived susceptibility (two items), perceived severity (two items), perceived barriers (two items), perceived benefits (two items), cues to action (two items), and self-efficacy (two items). Participants were asked to indicate their level of agreement with health beliefs toward pregnancy planning on a 5-point scale ranging from 1 (strongly disagree) to 5 (strongly agree). The total score for each factor ranged from 2 to 10 (one item of self-efficacy was reverse-coded). The construct validity of these scales in the present study was confirmed by factor analysis, with a Kaiser-Meyer-Olkin (KMO) score of 0.78 and satisfactory results on Bartlett’s test of sphericit (χ^2^ = 2940.79, *p* < 0.001). The reliability was confirmed by Cronbach’s α values of 0.70, 0.67, 0.64, 0.65, 0.65, and 0.77, respectively. The total Cronbach’s α was 0.76.

#### 2.3.2. Fertility Knowledge

Fertility knowledge was assessed using the 13-item questionnaire utilized by Bunting et al. [22]. The instrument, reported to have satisfactory face validity and reliability, was translated into Korean. This scale included three domains: (i) indicators of reduced fertility, (ii) misconceptions about fertility, and (iii) basic facts about infertility. The response scale was “true,” “false,” or “don’t know.” A correct answer was assigned 1 point and an incorrect or “don’t know” answer 0 points, and the total score ranged from 0 to 13. The reliability of the fertility knowledge scale in the previous study was confirmed by Cronbach’s α values of 0.79 in most countries (except for Italy [0.59] and Turkey [0.41]), and Cronbach’s α in the present study was 0.60.

#### 2.3.3. General Characteristics

Participants stated their age in years, grade, gender, whether they were religious (yes/no), and economic status (low/medium/high). They also indicated whether they had smoked in the previous month (yes/no), whether they had drunk alcohol in the previous month (yes/no), how stressed they were (not at all/a little/much/very much), how depressed they were (not at all/a little/very much), whether they had thought about suicide (yes/no), whether they had sexual experience (yes/no), how often they practiced contraception (not at all/a little/much/very much), and whether they had experiences of pregnancy (yes/no). Based on previous studies [14,15,16,17] that confirmed associations between pregnancy planning and reproductive health–promoting behaviors, some of the general characteristics (age, income, smoking, alcohol drinking, feeling stressed, feeling depressed, and sexual experience) were included in step 1 of the hierarchical regression analysis.

### 2.4. Data Analysis

Descriptive analyses were performed using frequencies (percentages) for categorical variables and means (standard deviation) for continuous variables. The chi-square test, Fisher’s exact test, and independent t-test were used to analyze differences in sociodemographic and health-related characteristics, fertility knowledge, and health beliefs regarding pregnancy planning between women and men. Hierarchical regression was used to examine factors influencing self-efficacy toward pregnancy planning. Independent variables were entered into the equation in a three-step order specified by the researcher (step 1: general characteristics; step 2: fertility knowledge; step 3: health belief variables). For all statistical tests, a two-tailed p-value < 0.05 was considered to indicate statistical significance. All statistical analyses were performed using Stata version 16 (StataCorp, College Station, TX, USA) and SPSS version 24 (IBM Corp., Armonk, NY, USA).

## 3. Results

### 3.1. General and Fertility-Related Characteristics of Participants

A total of 819 students answered the questionnaire, of whom 476 were women and 343 were men. Participants’ general characteristics and fertility-related characteristics are presented in Table 1. The mean age was 23.64 (± 2.35) years for men and 22.60 (± 1.94) years for women. Most respondents were not religious (70.8%). The general characteristics that showed a statistically significant difference according to gender were age (t = 6.96, *p* < 0.001), economic status (χ^2^ = 6.71, *p* = 0.035), smoking (χ^2^ = 12.82, *p* < 0.001), alcohol drinking (χ^2^ = 13.38, *p* = 0.001), feeling stressed (χ^2^ = 9.60, *p* = 0.022), feeling depressed (χ^2^ = 9.23, *p* = 0.010), and having thoughts about suicide (χ^2^ = 11.54, *p* < 0.001).

Most participants answered that pregnancy planning is necessary (88.6%). Furthermore, sexual experience was a fertility-related characteristic with a statistically significant difference between men and women (χ^2^ = 27.48, *p* < 0.001).

### 3.2. Fertility Knowledge

The percentage of correct answers for fertility knowledge ranged from 15.4% to 91.8% among all participants (Table 2). The item with the highest correct answer rate for both men and women was “smoking decreases female fertility,” which was answered correctly by 88.9% of women and 95.9% of men. The item with the lowest correct answer rate was “if a man has mumps after puberty, he is more likely to have fertility problems later,” with a correct response rate of 13.5% for women and 18.1% for men. The male students had a higher level of fertility knowledge than female students (t = 2.06, *p* = 0.040).

### 3.3. Health Belief Variables about Pregnancy Planning and Self-Efficacy toward Pregnancy Planning

Comparisons of health belief variables related to pregnancy planning and self-efficacy toward pregnancy planning by gender are shown in Table 3. Male students indicated a greater perceived benefit of pregnancy planning than female students (t = 4.18, *p* < 0.001), whereas female students had a higher score regarding perceived barriers of pregnancy planning than male students (t = 4.18, *p* < 0.001). Additionally, male students had higher self-efficacy toward pregnancy planning than female students (t = 3.07, *p* = 0.002).

### 3.4. Factors Associated with Self-Efficacy toward Pregnancy Planning

Table 4 describes the factors influencing self-efficacy toward pregnancy planning. General characteristics were entered in step 1. The general characteristics with a significant influence on self-efficacy toward pregnancy planning in female students were smoking (β = −0.10, *p* = 0.031) and feeling depressed (β = −0.17, *p* = 0.002), whereas the corresponding variables in male students were alcohol drinking (β = −0.12, *p* = 0.025) and feeling stressed (β = −0.14, *p* = 0.026). Scores on fertility knowledge were entered in step 2. In step 2, the significant factors in female students were smoking (β = −0.10, *p* = 0.036), feeling depressed (β = −0.16, *p* = 0.003), and fertility knowledge (β = 0.11, *p* = 0.019), whereas the corresponding predictors in male students were alcohol drinking (β = −0.12, *p* = 0.025) and feeling stressed (β = −0.14, *p* = 0.026). In step 3, when health belief variables were included, the full model was significant (F = 26.53, *p* < 0.001; F = 14.93, *p* < 0.001 for men and women, respectively). The final model was a significantly better predictor than the model in step 2. In step 3, the factors with a significant influence on self-efficacy toward pregnancy planning in female students were feeling depressed (β = −0.09, *p* = 0.030), fertility knowledge (β = 0.08, *p* = 0.025), perceived barriers (β = −0.57, *p* < 0.001), and cues to action (β = 0.16, *p* < 0.001), whereas the corresponding factors in male students were perceived benefits (β = 0.12, *p* = 0.020), perceived barriers (β = −0.44, *p* < 0.001), and cues to action (β = 0.16, *p* = 0.001).

## 4. Discussion

The purpose of this study was to explore Korean college students’ thoughts about pregnancy planning under the assumption that pregnancy planning may start early (i.e., before marriage or pregnancy). In this study, we confirmed the existence of gender differences in self-efficacy toward pregnancy planning and the factors influencing self-efficacy toward pregnancy planning.

First, female students had lower self-efficacy toward pregnancy planning than male students. According to previous studies, Korean female college students not only felt that childbirth and child-rearing were more burdensome than male students [23] but also had negative attitudes toward marriage and childbirth [24,25]. In the context of those findings on negative perceptions about marriage, childbirth, and child rearing, a possible explanation for the result of the present study that female college students expressed lower self-efficacy toward pregnancy planning than male students is that female college students focused on studying or self-realization, not on pregnancy or childbirth.

Regarding the factors influencing self-efficacy toward pregnancy planning in men and women, the factor with the strongest influence was perceived barriers, followed by cues to action. Therefore, in order to increase self-efficacy toward pregnancy planning among unmarried male and female college students, it would be helpful to reduce perceived barriers by avoiding the perception that pregnancy planning is annoying or difficult. In addition, the impact of family and friends should be considered when planning measures to improve college students’ self-efficacy toward pregnancy planning, because Korean university students are influenced by their family members [26], and their sexual attitudes and behavior are particularly strongly influenced by their friends [27].

Of the health belief variables that were analyzed, perceived benefits only had an impact on self-efficacy toward pregnancy planning among men. Emphasizing the benefits of pregnancy planning on men may therefore increase their self-efficacy toward pregnancy planning. In contrast, female students with a lower level of depression and greater fertility knowledge expressed higher self-efficacy toward pregnancy planning. These results appear to be somewhat similar to those of a previous study [28], which reported that women with depression were more likely to have unplanned pregnancies, although differences between the participants in these two studies hinder the direct comparability of their results. Therefore, it is especially necessary to raise awareness of pregnancy planning among women with a higher level of depression. It is also important to improve fertility knowledge among female students in order to increase their self-efficacy toward pregnancy planning. The American College of Obstetricians and Gynecologists has emphasized that women should optimize their health and knowledge before planning for pregnancy to achieve healthy pregnancy outcomes [29].

With regard to fertility knowledge, it was noteworthy that neither men nor women were generally aware that mumps after puberty affects future reproductive capacity. Thus, college students need to be educated on this fact regarding mumps. In this study, male students had higher fertility knowledge scores than female students. In contrast, previous studies measuring fertility knowledge reported that men lacked fertility knowledge relative to women [30,31]. Further research needs to confirm these gender differences in fertility knowledge. Regarding the fertility knowledge scale, we carried out the entire process of translation of the original version, including back-translation by a professional. Nonetheless, it will be necessary to confirm the reliability and validity of the scale in the future study.

Since the data collection was done at a single university, a limitation of this study is that the results may not be generalizable to all Korean college students. Although self-efficacy toward pregnancy planning has implications for the future, it is distinct from assessing whether unmarried college students actually engage in pregnancy planning. Future follow-up research is needed to determine whether self-efficacy is associated with actual pregnancy planning for prospective couples. Despite these limitations, this study is the first to focus on self-efficacy toward pregnancy planning and factors influencing self-efficacy toward pregnancy planning among unmarried college students, from the perspective of pregnancy planning as the initial step of preconception care. Our study is meaningful in that it may have provided college students with a chance to consider pregnancy planning before conception, which may promote healthy future pregnancies and childbirths. Moreover, the HBM was confirmed as suitable conceptual framework for predicting self-efficacy toward pregnancy planning.

## 5. Conclusions

Our study showed that gender differences were present in self-efficacy toward pregnancy planning, and that the factors influencing self-efficacy toward pregnancy planning had both similarities and differences between men and women. Therefore, the following strategies are proposed. First, both men and women should identify relevant barriers and cues to action and take steps to resolve them. Second, the benefits of pregnancy planning should be highlighted for men, while for women, steps should be taken to improve fertility knowledge and to pay more attention to women with depression. Therefore, gender differences should be considered when developing interventions to promote self-efficacy toward pregnancy planning among college students.

## Figures and Tables

**Table 1 ijerph-17-03735-t001:** General and fertility-related characteristics by gender (*n* = 819).

Variables	Classification	Total	Women (476)	Men (343)	95% Confidence Interval of the Difference		χ^2^ or t (*p*)	Effect Size
		*n* (%) or M ± SD			Lower	Upper		V or d or φ
General characteristics								
Age (years)		23.04 ± 2.18	22.60 ± 1.94	23.64 ± 2.35	22.89	23.19	6.96 (< 0.001)	0.49 ^‡^
	18−24	613 (74.9)	401 (84.2)	212 (61.8)				
	25–29	206 (25.1)	75 (15.8)	131 (38.2)				
Grade	Freshman	108 (13.2)	65 (13.7)	43 (12.5)	10.9	15.5	1.73 (0.785)	0.05 ^∮^
	Sophomore	142 (17.3)	80 (16.8)	62 (18.1)	14.7	19.9		
	Junior	205 (25.0)	114 (24.0)	91 (26.5)	22.0	28.0		
	Senior	319 (39.0)	188 (39.5)	131 (38.2)	35.7	42.3		
	Other	45 (5.5)	29 (6.1)	16 (4.7)	3.9	7.1		
Being religious	Yes	239 (29.2)	137 (28.8)	102 (29.7)	26.1	32.3	0.09 (0.767)	0.01
Economic status	Low	50 (6.1)	22 (4.6)	28 (8.2)	4.5	7.7	6.71 (0.035)	0.09 ^∮^
	Medium	528 (64.5)	302 (43.5)	226 (65.9)	61.2	67.8		
	High	241 (29.4)	152 (31.9)	89 (25.9)	26.3	32.5		
Smoking	Yes	91 (11.1)	37 (7.8)	54 (15.7)	8.9	13.3	12.82 (< 0.001)	0.13
Alcohol drinking	Yes	639 (78.0)	350 (73.5)	289 (84.3)	75.2	80.8	13.38 (< 0.001)	0.13
Feeling stressed	Not at all	38 (4.6)	16 (3.4)	22 (6.4)	3.2	6.0	9.60 (0.022)	0.11 ^∮^
	A little	451 (55.1)	250 (52.5)	201 (58.6)	51.7	58.5		
	A lot	280 (34.2)	177 (37.2)	103 (30.0)	31.0	37.4		
	Very much	50 (6.1)	33 (6.9)	17 (5.0)	4.5	7.7		
Feeling depressed	Not at all	432 (52.8)	230 (48.3)	202 (58.9)	49.4	56.2	9.23 (0.010)	0.11 ^∮^
	A little	363 (44.3)	232 (48.7)	131 (38.2)	40.9	47.7		
	Very much	24 (2.9)	14 (3.0)	10 (2.9)	1.8	4.0		
Have thoughts about suicide	Yes	197 (24.0)	135 (28.4)	62 (18.1)	21.1	26.9	11.54 (0.001)	0.12
Fertility-related characteristics								
Sexual experience	Yes	442 (54.0)	220 (46.2)	222 (64.7)	50.6	57.4	27.48 (< 0.001)	0.18
Contraception use	Never	12 (2.7)	9 (4.1)	3 (1.4)	1.6	3.8	5.24 (0.155)	0.11 ^∮^
(442)	Sometimes	23 (5.2)	13 (5.9)	10 (4.5)	3.7	6.7		
	Often	125 (28.3)	55 (25.0)	70 (31.5)	25.2	31.4		
	Always	282 (63.8)	143 (56.0)	139 (62.6)	60.5	67.1		
Pregnancy experience (442)	Yes	5 (1.1)	4 (1.8)	1 (0.4)	0.4	1.8	1.85 ^†^ (0.215)	0.06
Pregnancy planning is necessary	Yes	726 (88.6)	428 (89.9)	298 (86.9)	86.4	90.8	1.83 (0.177)	0.05

Note: ^†^—Fisher’s exact test; ^‡^—Cohen’s d; ^∮^—Cramer’s V; SD—Standard deviation.

**Table 2 ijerph-17-03735-t002:** Fertility knowledge by gender (*n* = 819).

Question	Classification	Total	Women (476)	Men (343)	95% Confidence Interval of the Difference		χ^2^ or t (*p*)	Effect Size
		*n* (%) or M ± SD			Lower	Upper		V or d
A woman is less fertile after the age of 36 years.	FALSE	78 (9.5)	58 (12.2)	20 (5.8)	7.5	11.5	9.87 (0.007)	0.11
	TRUE	625 (76.3)	349 (73.3)	276 (80.5)	73.4	79.2		
	DON’T KNOW	116 (14.2)	69 (14.5)	47 (13.7)	11.8	16.6		
A couple is classified as infertile if they do not achieve pregnancy after one year of regular sexual intercourse (without using contraception).	FALSE	122 (14.9)	78 (16.4)	44 (12.8)	12.5	17.3	12.31 (0.002)	0.12
	TRUE	445 (54.3)	234 (49.2)	211 (61.5)	50.9	57.7		
	DON’T KNOW	252 (30.8)	164 (34.4)	88 (25.7)	27.6	34.0		
Smoking decreases female fertility.	FALSE	48 (5.9)	39 (8.2)	9 (2.6)	4.3	7.5	18.09 (< 0.001)	0.15
	TRUE	701 (85.6)	387 (81.3)	314 (91.6)	83.2	88.0		
	DON’T KNOW	70 (8.5)	50 (10.5)	20 (5.8)	6.6	10.4		
Smoking decreases male fertility.	FALSE	18 (2.2)	14 (2.9)	4 (1.2)	1.2	3.2	13.22 (0.001)	0.13
	TRUE	752 (91.8)	423 (88.9)	329 (95.9)	89.9	93.7		
	DON’T KNOW	49 (6.0)	39 (8.2)	10 (2.9)	4.4	7.6		
About 1 in 10 couples are infertile.	FALSE	48 (5.9)	25 (5.3)	23 (6.7)	4.3	7.5	0.95(0.620)	0.03
	TRUE	330 (40.3)	196 (41.2)	134 (39.1)	36.9	43.7		
	DON’T KNOW	441 (53.9)	255 (53.6)	186 (54.2)	50.5	57.3		
If a man produces sperm he is fertile.	FALSE	608 (74.2)	364 (76.5)	244 (71.1)	71.2	77.2	4.46 (0.108)	0.07
	TRUE	136 (16.6)	68 (14.3)	68 (19.8)	14.1	19.1		
	DON’T KNOW	75 (9.2)	44 (9.2)	31 (9.1)	7.2	11.2		
These days a woman in her 40s has a similar chance of getting pregnant as a woman in her 30 s.	FALSE	371 (45.3)	194 (40.7)	177 (51.6)	41.9	48.7	10.22 (0.006)	0.11
	TRUE	116 (14.2)	77 (16.2)	39 (11.4)	11.8	16.6		
	DON’T KNOW	332 (40.5)	205 (43.1)	127 (37.0)	37.1	43.9		
Having a healthy lifestyle makes you fertile.	FALSE	148 (18.1)	101 (21.2)	47 (13.7)	15.5	20.7	8.16 (0.017)	0.10
	TRUE	561 (68.5)	310 (65.1)	251 (73.2)	65.3	71.7		
	DON’T KNOW	110 (13.4)	65 (13.7)	45 (13.1)	11.1	15.7		
If a man has mumps after puberty, he is more likely to have fertility problems later.	FALSE	91 (11.1)	45 (9.5)	46 (13.4)	8.9	13.3	9.88 (0.007)	0.10
	TRUE	126 (15.4)	64 (13.5)	62 (18.1)	12.9	17.9		
	DON’T KNOW	602 (73.5)	367 (77.0)	235 (68.5)	70.5	76.5		
A woman who never menstruates is still fertile.	FALSE	467 (57.0)	306 (64.3)	161 (46.9)	53.6	60.4	25.80 (< 0.001)	0.18
	TRUE	120 (14.7)	63 (13.2)	57 (16.6)	12.3	17.1		
	DON’T KNOW	232 (28.3)	107 (22.5)	125 (36.5)	25.2	31.4		
If a woman is overweight by more than two stone (13 kg or 28 pounds) then she may not be able to get pregnant.	FALSE	283 (34.6)	160 (33.6)	123 (35.9)	31.3	37.9	2.34 (0.311)	0.05
	TRUE	226 (27.6)	141 (29.6)	85 (24.8)	24.5	30.7		
	DON’T KNOW	310 (37.8)	175 (36.8)	135 (39.3)	34.5	41.1		
If a man can achieve an erection, that is an indication that he is fertile.	FALSE	705 (86.1)	407 (85.5)	298 (86.9)	83.7	88.5	6.01 (0.050)	0.09
	TRUE	27 (3.3)	11 (2.3)	16 (4.7)	2.1	4.5		
	DON’T KNOW	87 (10.6)	58 (12.2)	29 (8.5)	8.5	12.7		
People who have had a sexually transmitted disease are likely to have reduced fertility.	FALSE	148 (18.1)	90 (18.9)	58 (16.9)	15.5	20.7	14.70 (0.001)	0.13
	TRUE	411 (50.2)	213 (44.8)	198 (57.7)	46.8	53.6		
	DON’T KNOW	260 (31.7)	173 (36.3)	87 (25.4)	28.5	34.9		
Total score		7.22 ± 2.03	7.10 ± 2.05	7.39 ± 1.99	7.08	7.36	2.06 (0.040)	0.15 ^†^

Note: ^†^—Cohen’s d; SD—Standard deviation; Underlined text means correct answer.

**Table 3 ijerph-17-03735-t003:** Health beliefs related to pregnancy planning and self-efficacy toward pregnancy planning by gender (*n* = 819).

Health Belief Variables	Total	Women (476)	Men (343)	t (*p*)
	M ± SD			
1. Perceived susceptibility				
1) If a pregnancy is unplanned, the baby is more likely to be born with health problems.	3.00 ± 1.05	2.96 ± 1.05	3.06 ± 1.05	1.26 (0.207)
2) Unplanned pregnancies place pregnant women’s health at risk.	3.50 ± 0.98	3.56 ± 0.98	3.42 ± 0.99	−2.01 (0.045)
Subtotal	6.50 ± 1.79	6.52 ± 1.75	6.48 ± 1.84	−0.36 (0.718)
2. Perceived severity				
1) If a pregnancy is unplanned, the risk of miscarriage increases.	3.70 ± 0.90	3.74 ± 0.85	3.66 ± 0.95	−1.25 (0.212)
2) If a pregnancy is unplanned, the baby’s health could be at risk.	3.01 ± 0.95	2.98 ± 0.95	3.05 ± 0.94	0.95 (0.345)
Subtotal	6.71 ± 1.60	6.72 ± 1.56	6.70 ± 1.65	−0.14 (0.889)
3. Perceived benefits				
1) If you plan for pregnancy, the baby is more likely to be born healthy.	3.34 ± 0.89	3.26 ± 0.89	3.44 ± 0.89	2.94 (0.003)
2) If you plan for pregnancy, the mother will be healthy because of sufficient preparation.	3.80 ± 0.83	3.69 ± 0.92	3.96 ± 0.66	4.66 (< 0.001)
Subtotal	7.14 ± 1.58	6.95 ± 1.56	7.40 ± 1.31	4.39 (< 0.001)
4. Perceived barriers				
1) It will be hard for me to plan for pregnancy.	2.46 ± 1.06	2.59 ± 1.10	2.27 ± 0.99	−4.35 (< 0.001)
2) It will be bothersome to receive preconception care.	2.32 ± 1.17	2.47 ± 1.23	2.10 ± 1.06	−4.47 (< 0.001)
Subtotal	4.78 ± 1.93	5.07 ± 2.00	4.37 ± 1.74	−5.15 (< 0.001)
5. Cues to action				
1) My family will advise me to plan for pregnancy.	3.88 ± 0.87	3.87 ± 0.91	3.89 ± 0.83	0.43 (0.668)
2) My friends will help me to plan for pregnancy.	3.36 ± 0.91	3.38 ± 0.90	3.32 ± 0.92	−1.00 (0.315)
Subtotal	7.23 ± 1.53	7.25 ± 1.56	7.21 ± 1.50	−0.35 (0.727)
6. Self-efficacy toward pregnancy planning				
1. I’m not sure if I can plan to get pregnant. ^†^	2.68 ± 1.05	2.78 ± 1.11	2.54 ± 0.94	−3.35 (0.001)
2. I can plan to get pregnant.	3.65 ± 0.91	3.60 ± 0.95	3.73 ± 0.84	2.11 (0.035)
Subtotal	6.97 ± 1.77	6.81 ± 1.87	7.20 ± 1.59	3.07 (0.002)
Kaiser-Meyer-Olki 0.78; Bartlett’s test = 2940.79; *p* < 0.001; cumulative variance = 78.32; total Cronbach’s α = 0.76				

Note: ^†^—The response was summed after reverse-coding the negative item; SD—Standard deviation.

**Table 4 ijerph-17-03735-t004:** Factors influencing self-efficacy toward pregnancy planning by gender (*n* = 819).

Independent Variables	Women (476)					
	Step 1		Step 2		Step 3	
	B	β (*p*)	B	β (*p*)	B	β (*p*)
(Constant)	8.16		7.62		8.08	
Age	−0.07	−0.07 (0.145)	−0.07	−0.07 (0.125)	−0.06	−0.06 (0.118)
Economic status	0.26	0.08 (0.102)	0.26	0.07 (0.105)	0.17	0.05 (0.169)
Smoking^†^	−0.69	−0.10 (0.031)	−0.67	−0.10 (0.036)	0.17	0.02 (0.509)
Alcohol drinking^†^	0.19	0.05 (0.322)	0.15	0.04 (0.439)	−0.07	−0.02 (0.640)
Feeling stressed	0.14	0.05 (0.338)	0.12	0.04 (0.409)	0.22	0.08 (0.058)
Feeling depressed	−0.56	−0.17 (0.002)	−0.55	−0.16 (0.003)	−0.31	−0.09 (0.030)
Sexual experience^†^	−0.04	−0.01 (0.826)	−0.04	−0.01 (0.810)	0.09	0.02 (0.524)
Fertility knowledge			0.10	0.11 (0.019)	0.07	0.08 (0.025)
Perceived susceptibility					−0.04	−0.04 (0.371)
Perceived severity					0.05	0.05 (0.328)
Benefits					0.04	0.03 (0.370)
Barriers					−0.53	−0.57 (< 0.001)
Cues to action					0.19	0.16 (< 0.001)
R^2^ (Δ R^2^)	0.05		0.06 (0.01)		0.43 (0.37)	
*Adj* R^2^	0.04		0.05		0.41	
F (*p*)	3.61 (0.001)		3.89 (< 0.001)		26.53 (< 0.001)	
**Independent Variables**	**Men (343)**					
	**Step 1**		**Step 2**		**Step 3**	
	**B**	**β (*p*)**	**B**	**β (*p*)**	**B**	**β (*p*)**
(Constant)	8.44		8.42		7.92	
Age	−0.01	−0.01 (0.854)	−0.01	−0.01 (0.852)	−0.04	−0.06 (0.236)
Economic status	0.05	0.02 (0.749)	0.05	0.02 (0.753)	0.08	0.03 (0.534)
Smoking ^†^	−0.05	−0.01 (0.822)	−0.05	−0.01 (0.823)	−0.16	−0.04 (0.408)
Alcohol drinking ^†^	−0.53	−0.12 (0.025)	−0.53	−0.12 (0.025)	−0.22	−0.05 (0.275)
Feeling stressed	−0.34	−0.14 (0.026)	−0.34	−0.14 (0.026)	−0.23	−0.10 (0.069)
Feeling depressed	−0.12	−0.04 (0.492)	−0.12	−0.04 (0.495)	−0.13	−0.04 (0.396)
Sexual experience ^†^	0.36	0.11 (0.061)	0.36	0.11 (0.062)	0.26	0.08 (0.113)
Fertility knowledge			0.00	0.01 (0.933)	−0.03	−0.04 (0.373)
Perceived susceptibility					−0.03	−0.03 (0.625)
Perceived severity					0.09	0.09 (0.129)
Benefits					0.15	0.12 (0.020)
Barriers					−0.40	−0.44 (< 0.001)
Cues to action					0.17	0.16 (0.001)
R^2^ (Δ R^2^)	0.06		0.06 (0.00)		0.37 (0.31)	
*Adj* R^2^	0.04		0.04		0.35	
F (*p*)	2.94 (0.005)		2.56 (0.010)		14.93 (< 0.001)	

Note: ^†^—Dummy variables (Ref. no).

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
