# Peer review of "Gender Differences in Factors Influencing Self-Efficacy Toward Pregnancy Planning among College Students in Korea"

_ijerph, 2020, doi:10.3390/ijerph17103735_

Round 1

Reviewer 1 Report

Kang SY and Kim HW conducted research work on determining the associations between individual factors and self-efficacy toward pregnancy planning among college students from one university in Korea. This is a cross-sectional study based on data from an online questionnaire. This study enrolled 1,273 unmarried students aged between 18 and 30. With the exclusion of missing data, the study population of this research included 819 students, with 476 women and 343 men. The authors first showed the demography, fertility-related information, fertility knowledge, health belief variables, and self-efficacy toward pregnancy planning between sexes among participated students. Secondly, the authors used hierarchical linear regression analysis to examine the associations of individual factors with self-efficacy toward pregnancy planning in men and women. Overall, this is a well-written manuscript describing the differential influences of individual factors on self-efficacy toward pregnancy planning between male and female college students in Korea. The research outcome of this study highlights the importance of the pilot work in developing future preconception care among college students in Korea. I have a few question and suggestions as listed below,

1) According to AMA style, “sex” refers to the state of biologically being male or female; “gender” is a term that should be reserved for psychological and sociological context. Please edit your title and content of this manuscript as appropriate.

2) In Result 3.4, the authors used Step 1-3 in the text, but Model 1-3 in Table 4. Please make sure the description is consistent throughout the article.

3) It is interesting to see the health belief has a stronger association with the self-efficacy toward pregnancy planning than demography and fertility knowledge in both men and women based on the comparisons between regression models. Can the authors perform statistical analysis to compare the associations between sex using the setting of model 3?

Reviewer 2 Report

I congratulate the authors on a well conducted study and a well-written paper which will no doubt be of interest to many ijerph readers-

However I also have some important issues needing clarification/adjustment:

You have followed up on the incomplete responders, naturally a question arises as to whether those who did not fully complete the questionniare also have different attitudes/self-efficacy etc to the completers- your sample of incompletes is only 84- what about the other 300 or so subjects? why are they not inlcuded in the incomplete analysis?

2.3.2 Fertility knowledge- it seems important here to comment on the divergent cronbach alpha scores between countries- is there any explanation offered as to why there are values from 0.41 to 0.79? your own lies between these at 0.60 is it language or something else?

Minor issues/typographical errors

Line 59  delete: 'still' so the sentence now begins with the word 'Traditionally'

Line 142 has a typo you have written women twice where one of the values is for men.

Line 146 change 'thought' for 'thoughts'

Table 2 needs some attention- the: False, True, Don't know categories do not line up properly. 

Table1 & 2 my feeling is that for your values reported here you should also include an effect size and 95% confidence intervals.
